# Cancer stem cell-derived extracellular vesicles preferentially target MHC-II–macrophages and PD1+ T cells in the tumor microenvironment

**Patricia Gonzalez-Callejo**[1,2], **Zihan Guo**[1,3], **Tahereh Ziglari**[1], **Natalie Marcia Claudio**[1,4], **Kayla Hoang Nguyen**[1,3], **Naoki Oshimori**[1,3,4,5], **Joaquim Seras-Franzoso**[6], **Ferdinando Pucci**[1,3,4]*

1 Department of Cell, Developmental & Cancer Biology, Oregon Health and Science University, Portland, Oregon, United States of America, 2 Bionanoplasmonics Group, CIC biomaGUNE, Donostia-San Sebastián, Spain, 3 Program in Biomedical Sciences, Oregon Health and Science University, Portland, Oregon, United States of America, 4 Department of Otolaryngology-Head and Neck Surgery, Oregon Health and Science University, Portland, Oregon, United States of America, 5 Department of Dermatology, Oregon Health and Science University, Portland, Oregon, United States of America, 6 Drug Delivery and Targeting Group, Vall d'Hebron Research Institute (VHIR), Barcelona, Spain

* pucci@ohsu.edu

**Data Availability Statement:** All relevant data are within the paper and its Supporting Information files.

## Abstract

Immunotherapy is an approved treatment option for head and neck squamous cell carcinoma (HNSCC). However, the response rate to immune checkpoint blockade is only 13% for recurrent HNSCC, highlighting the urgent need to better understand tumor-immune interplay, with the ultimate goal of improving patient outcomes. HNSCC present high local recurrence rates and therapy resistance that can be attributed to the presence of cancer stem cells (CSC) within tumors. CSC exhibit singular properties that enable them to avoid immune detection and eradication. How CSC communicate with immune cells and which immune cell types are preferentially found within the CSC niche are still open questions. Here, we used genetic approaches to specifically label CSC-derived extracellular vesicles (EVs) and to perform Sortase-mediated *in vivo* proximity labeling of CSC niche cells. We identified specific immune cell subsets that were selectively targeted by EV[CSC] and that were found in the CSC niche. Native EV[CSC] preferentially targeted MHC-II–macrophages and PD1+ T cells in the tumor microenvironment, which were the same immune cell subsets enriched within the CSC niche. These observations indicate that the use of genetic technologies able to track EVs without *in vitro* isolation are a valuable tool to unveil the biology of native EV[CSC].

## Introduction

Head and neck squamous cell carcinoma (HNSCC) accounts for approximately 90% of oral and oropharyngeal cancer with over 400,000 new cases and more than 150,000 deaths reported

**Funding:** European Molecular Biology Organization (EMBO):Patricia Gonzalez-Callejo short-term fellowship; V Foundation for Cancer Research (VFCR):Natalie M Claudio,Ferdinando Pucci 2019-012.

**Competing interests:** The authors have declared that no competing interests exist.

each year worldwide [1]. Advances in traditional treatments (surgery, radiotherapy, chemotherapy) have failed to increase survival due to patients presenting incurable advanced-stage disease and lymph node metastasis that ultimately cause their death [2].

Immunotherapy is an approved treatment option for Head and neck squamous cell carcinoma (HNSCC) [3]. However, the response rate to immune checkpoint blockade is only 13% for recurrent HNSCC, highlighting the urgent need to better understand tumor-immune interplay, with the ultimate goal of improving patient outcomes [3].

HNSCC present high local recurrence rates and therapy resistance that can be attributed to the presence of Cancer Stem Cells (CSC) within tumors. Several cell biomarkers such as CD44 [4–6], CD133 [7, 8], SMAD Responsive Element (SRE) [9] and Aldehyde Dehydrogenase (ALDH) activity [10–13] have identified specific CSC-like populations in HNSCC tumors with enhanced tumorigenic potential and resistance to chemo- or radiotherapy. CSC exhibit unique malignant intrinsic characteristics and play key roles in tumor initiation, growth and metastasis. CSC are also believed to drive therapy resistance and tumor relapse, as they can survive and dynamically adapt to changing and unfavorable environmental conditions [14–19].

CSC exhibit singular properties that enable them to avoid immune detection and eradication [20]. Recently, a number of studies have shown that CSC contribute to the generation of an immunosuppressive, pro-tumorigenic immune milieu by regulating the activity of various immune cells in the tumor microenvironment (TME). CSC can modulate T cells, tumor-associated macrophages (TAMs) and myeloid-derived suppressor cells activity towards immunosuppressive pathways [20–25]. Importantly, these immune cells can also sustain CSC stemness and survival [25–28]. Such complex communication network between CSC and immune cells operates through various secreted cytokines, chemokines, growth factors and proteins of the extracellular matrix (ECM) [25–29]. Whether extracellular vesicles (EVs) also play a role is still unknown.

Emerging evidence has shown that tumors can interfere with host immunity by secreting EVs [30]. EVs are defined as a heterogeneous collection of lipid bilayer membrane-enclosed vesicles naturally secreted by both prokaryotic and eukaryotic cells and that carry a complex cargo of mRNAs, lipids, metabolites, proteins and non-coding RNAs able to induce a response when signaling to EV-recipient cells [31–39]. EVs of varying size, biogenesis and cargo content can be released from a single cell. Moreover, EV secretion pattern and content can change with changes in the physiological state of the parental cell [40, 41]. Once released, EVs can interact with cells in the immediate vicinity or at distant locations via transfer through lymphatic and blood circulation. Tumor derived EVs (tEVs) can affect the proliferation, apoptosis, cytokine production and reprogramming of both innate and adaptive immune cells, thereby modifying anti-cancer immune responses [42–49]. Whether these functions belong to specific subpopulation of tEVs, such as those released by CSC, is still unclear.

## Materials and methods

### Cell lines and culture conditions

**Parental cell lines.** Murine oral squamous cell carcinoma (OSCC) cell lines MOC2 (a chemical carcinogenesis model) and mEER (a Ras-dependent, HPV16-E6/E7-dependent model) were obtained from Kerafast, Inc, and Dr. Varner (UCSD), respectively. Both cell lines were routinely maintained in IMDM/DMEM/F12 (50:25:25) supplemented with 5% fetal bovine serum (FBS), 2 mM L-Glutamine, 1x Pen/Strep solution, Hydrocortizone (25ug/uL), Cholera Toxin (0.25ug/uL), Transferrin (25ug/uL), Insulin (10ug/uL), Tri-Iodo-Thyronine (0.2ug/uL), E.G.F. (10ug/mL). All cell cultures were propagated at 37˚C and 5% $CO_2$ in a humidified incubator.

**Lentiviral vectors.** Detailed schemes of the lentivectors employed to tag EVs are presented in **S1 Fig**. A brief description of the function of each LV is detailed below.

- Lentiviral (LV) transfer plasmid coding for CD63-eGFP (and *dLNGFR)* was cloned as previously described [49] and used to tag total tEVs, as EVs marker CD63-eGFP expression is subjected to the regulation of the phosphoglycerate kinase (PGK) constitutive promoter.

- LV transfer plasmids coding for ***ALDH1A1:CD63-eGFP*** and ***SRE:CD63-eGFP*** expression cassettes were designed by us and cloned by Genewiz. Those LV were used to tag CSC derived EVs as CD63-eGFP expression is subjected to the regulation of CSC specific promoters *ALDH1A1* and *SRE* [9–13].

- LV transfer plasmid coding for ***ALDH1A1:SrtA*** was designed by us and cloned by Genewiz. This plasmid was used to engineer CSC and their EVs to display a membrane-bound form of Sortase A (SrtA), a bacterial transpeptidase that can catalyze the transfer of reporter molecules on the surface of EV-binding cells and cells interacting with CSC.

- LV transfer plasmid coding for ***SS-mSca-LPETGG:mCMV-PGK:CD63-eGFP*** was designed by us and cloned by Genewiz. This plasmid was used to tag total tEVs and to provide SrtA substrate (mScarlet red fluorescence protein fused to the SrtA recognition sequence LPETGG). A signal sequence (SS) instructs the cell to secrete mSca in the extracellular environment.

**Generation of modified cell lines expressing tEVs and tEVs$^{CSC}$ markers.** In order to genetically label EVs, mEER and MOC2 cell lines were transduced as previously reported by our group [49]. Briefly, LV transfer plasmids were propagated in Escherichia coli DH5α. Maxi-prep was performed with Endo-free Macherey-Nagel kit. Unconcentrated lentiviral vectors were generated. MOC2 and mEER cells were seeded at a concentration of $10^5$ cells per well in a 6-well plate and transduced with LV vector supernatants (1:1 ratio with complete media) in the presence of 1 μg/ml polybrene (Millipore). For the simultaneous *in vivo* detection of both tEVs$^{total}$ and tEVs$^{CSC}$, mEER cell line was consecutively transduced with *SS-mSca-LPETGG*:*mCMV-PGK*:*CD63-eGFP* and *ALDH1A1*:*SrtA* LV vectors in the presence of 1 μg/ml polybrene (Millipore). Engineered cell lines were cultured under the same conditions than the ones described for the parental cell lines and propagated at 37˚C and 5% $CO_2$ in a humidified incubator.

**CSC niche cell labeling strategy.** The strategy employed to detect CSC niche cells is based on detection of surface cell-cell contacts via SrtA, which we previously developed and extensively characterized [50]. Briefly, SrtA is a membrane-bound bacterial transpeptidase, which can catalyze the transfer of a reporter on nearby cells. SrtA is engineered to be present on the outer side of membranes (**S1C and S2A Figs)**. The reporter is monomeric Scarlett fluorescent protein (mSca) fused with SrtA recognition sequence (LPETGG) and with a secretory signal sequence that allows the release of mSca as an extracellular protein (**S1C Fig**). Thanks to its generic transmembrane domain, SrtA labels the cell surface as well as both CD63+ and CD81+ EVs (**S2B, S2C Fig**). When SrtA+ EVs and/or CSC come in close contact with other cells, and when the SrtA substrate mSca is present, SrtA covalently links mSca to any membrane protein with an N-terminal glycine (such as MHC-I, MHC-II, VE-Cadherin, CD19, integrins, and others) [50]. In order to control for internalization of mSca by neighbouring cells, we used a control group in which CSC do not express SrtA while the bulk of tumor cells (including CSC) still secrete mSca. The pan-tumor expression of CD63-eGFP served as internal control for total tumor-immune interactions. To establish baseline GFP fluorescence, we

used a control group lacking CD63-eGFP. The values for SrtA labelling were calculated by subtracting baseline internalization of mSca observed in the control group.

**Copy number assay.** Total DNA was extracted from genetically modified 200,000 mEER and MOC2 *ALDH1A1*:*CD63-eGFP* and *SRE*:*CD63-eGFP* cell lines using QIAamp DNA Micro Kit (Qiagen). LV sequence was detected using a custom taqman assay (Applied Biosystems) on RRE sequence in a Viia7 PCR system. TaqMan probes for reference genes were *ActinB*, *GusB* and *HPRT-1* (S1 Table). One copy per genome standard was used, as previously described [49].

**Orosphere formation assay.** Flow sorted 5000 mEER and MOC eGFP+ and eGFP- cells/ well were seeded in 6-well ultra-low attachment plates (Corner) in StemXVivo Serum-Free Tumorsphere Media (R&D Systems). Cells were cultured for 10 to 14 days, and orosphere formation efficiency was assessed manually counting the number of orospheres per well with diameters >50 $\mu$m for each condition using light microscopy.

**Stem gene profile validation.** Total RNA was extracted from 300,000 mEER and MOC2 (flow-sorted as *ALDH1A1*:*CD63-eGFP*+/- and *SRE*:*CD63-eGFP*+/- cells) using the RNeasy Mini Kit (Qiagen) and the RNA obtained was reverse transcribed using a HighCapacity cDNA Reverse Transcription Kit (Thermo Fisher Scientific) according to the manufacturer's instructions. The cDNA reverse transcription product was amplified with specific probes (S1 Table) by qPCR using TaqMan method (Thermo Fisher Scientific). The reaction was performed in triplicate on a Viia7 Real time PCR system (Applied Biosystems). Relative normalized quantities (NRQ) of mRNA expression were calculated using the comparative Ct method ($2^{-\Delta\Delta Ct}$) with two reference genes (GAPDH and Actin) used as endogenous controls with Excel software.

**Mice and tumor challenge.** Six- to eight-week-old C57BL/6J mice were purchased from Charles River Laboratories and housed under conventional conditions and provided with food and water ad libitum. Animal care was handled in accordance with the Guide for the Care and Use of Laboratory Animals of the Oregon Health & Science University (OHSU) and covered by OHSU Institutional Animal Care and Use Committee (IACUC). Euthanasia was performed in a $CO_2$ chamber.

For orthotopic tumor challenge, parental mEER cells (control) and genetically modified mEER *dLNGFR*:*mCMV-PGK*:*CD63-eGFP*, mEER *ALDH1A1*:*CD63-eGFP*, mEER *SRE*: *CD63-eGFP*, mEER *SS-mSca-LPETGG*:*mCMV-PGK*:*CD63-eGFP*, mEER *ALDH1A1*:*SrtA* and mEER *SS-mSca-LPETGG*:*mCMV-PGK*:*CD63-eGFP* plus *ALDH1A1*:*SrtA* cells were intradermally injected ($1x10^6$ in 50 μl of PBS) in the flank of six to eight week old C57BL/6J mice. After 12 days mice were euthanized and tumors collected, divided in two parts and processed either for flow cytometry or embedded in OCT compound for IF imaging analysis. Tumors were never allowed to grow bigger than 10mm in one dimension, as measured by caliper.

**Tumors IF imaging.** 5 μm thick OCT microsections from experimental tumors were mounted on glass slides for immunofluorescent labeling. Briefly, after 15' fixation with 4%PFA samples were washed in PBS-Tween 0.3% and primary antibodies, anti-rabbit eGFP (1:200, Abcam) anti-CD45 Biotin (1:200, Biolegend),anti-F4/80 Alexafluor-647 (1:200, Biolegend), anti-SOX2 (1:200 R&D) and anti-keratin 14 (1:200, Abcam) were supplemented in PBS/BSA 3% (w/v) and incubated O.N. at 4˚C. Samples were further washed 3 times in PBS-Tween 0.3% before the addition of secondary antibody. Goat Anti-rabbit AlexaFluor488 1:1000 and Streptavidin AlexaFluor568 1:500 were added and incubated 1h at RT. Slides were then washed and mounted with mounting media ProLong for visualization. Tumors were imaged using a Spinning Disk Confocal microscope (Yokogawa CSU-X1 on Zeiss Axio Observer). The measurement of the distance between eGFP+ cancer cells and F480+ Macrophages was made blindly

by 3 independent observers. This distance was quantitatively assessed with ImageJ software in 5 random tumor sections per mice group. Representative images were displayed.

**Flow cytometry.** Tumors were mechanically dissociated into single cell suspensions as previously described [49]. Cell suspensions were stained with conjugated antibodies (Biolegend, BD or eBiosciences) and Zombie aqua (Sigma). Following strategy was used to identify cells of interest:

- Tumor cells (CD63GFP+ Zombie aqua–CD45– CD31–)

- Endothelial cells (Zombie aqua–CD45– CD31+)

- B cells (Zombie aqua–CD45+ B220+)

- Macrophages MHC-II+(Zombie aqua–CD45+ CD11b+ F4/80+ II+)

- Macrophages MHC-II- (Zombie aqua–CD45+ CD11b+ F4/80+ II-)

- Inflammatory monocytes (Zombie aqua–CD45+ CD11b+ F4/80- CD11c+)

- Resident monocytes (Zombie aqua–CD45+ CD11b+ F4/80- CD11c-)

- Neutrophils (Zombie aqua–CD45+ F4/80- CD11c- SSChii)

- Dendritic cells (Zombie aqua–CD45+ CD11c+ F4/80– II+)

- PD-1 + T cells (Zombie aqua–CD45+ F4/80- B220- CD3+ PD-1+)

- PD-1—T cells (Zombie aqua–CD45+ F4/80- B220- CD3+ PD-1-)

Fluorochromes employed were the following: eGFP, Bv421, Bv605, Bv785, PE, PerCP, PC7, APC, A700, AC7.

Single EV analysis (**S2B, S2C Fig**) was performed as previously described [50]. Briefly, a 10-fold dilution of stock EV preparations in 0.2um filtered PBS was further serially diluted 2-fold to generate a 4-point dilution series. EV dilution series was stained with the following antibodies: CD63, CD81 and Flag, all from Biolegend and diluted to 0.1 ug/ml, without washing. Control unstained EV dilution series was also prepared. For each sample, we used a SSC trigger threshold value of 200 and the window extension at 3msec. EV staining samples were performed and analyzed in triplicate. After acquisition of 60ul (out of 200ul total), Triton-X100 was added to a final concentration of 0.5% (v/v) in each well and the plate was acquired again to confirm staining of EVs.

**Statistical analysis.** Bar graphs display mean value ± standard error of the mean (SEM). 2-way ANOVA Holm-Sidak´s test or non-parametric Tukey´s test were employed for multiple mean comparisons. The significance threshold was established at $p<0.05$, and significance levels were schematically assigned *$(0.01 \leq p < 0.05)$, **$(0.001 \leq p < 0.01)$, ***$(0.0001 \leq p,$ ****$(0.00001 \leq p)$. All the analyses and graphs were performed using GraphPad Prism 6 software (GraphPad, San Diego).

## Results

### Genetic labeling of cancer stem cell-derived extracellular vesicles

Tumor secreted EVs (tEVs) represent prominent regulators of the immune response in cancer [30, 42–49]. CSC secreted EVs (tEVs$^{CSC}$) are a subset of tEVs whose immunomodulating activity is still unknown. In order to start investigating whether tEVs$^{CSC}$ have a role in shaping immune cell activity in the TME, we genetically labeled tEVs$^{CSC}$ with fluorescent proteins. This approach allows to avoid any bias in EV composition due to *in vitro* isolation and assumptions on *in vivo* biodistribution of tEVs [49]. In particular, we genetically engineered

murine oral squamous cell carcinoma (OSCC) cell lines to express the vesicular membrane-associated protein CD63, fused with enhanced green fluorescence protein (CD63-eGFP) under the control of a CSC-specific promoter. We tested two different CSC-specific promoters, ALDH1A1 and SRE [9–13]. As reference controls, we genetically labeled the whole population of tEVs (including tEVs^CSC) by expressing the CD63-eGFP fusion protein under a constitutive promoter (PGK). We worked on two different OSCC cell lines, a chemical carcinogenesis model (MOC2) and a Ras-dependent, HPV16-E6/E7-dependent model (mEER). MOC2 carry the same mutations observed in human HN cancers, namely Trp53, MAPK and FAT whereas mEER+ have been engineered to express Hras(G12) and HPV-E6/E7. Together, the mutational landscape of these two cell lines model >95% of human pathology. As expected, the constitutive reporter (*PGK:CD63-eGFP+*) showed green fluorescence in virtually all tumor cells (**Fig 1A**). On the other hand, much less CD63-eGFP+ cells were observed in both MEER and MOC2 cells carrying the vectors *ALDH1A1:CD63-eGFP* (**Fig 1B**; **S3A Fig**) and *SRE:CD63-eGFP* (**Fig 1C**, **S3B Fig**). In order to confirm that the observed differential expression of the tEVs reporter CD63-eGFP was due to the restricted expression of the *ALDH1A1* and *SRE* promoters among CSC (and not because of low transduction efficiency), a lentiviral vector (LV) copy number assay was performed. These analyses showed that mEER *SRE:CD63-eGFP* cells carried on average 30 LV copies per cell (CpC), and that 5 CpC were detected in mEER *ALDH1A1:CD63-eGFP* cells, indicating full transduction of the tumor cell populations (**Fig 1D**). Flow cytometry analysis revealed that, expectedly, positive control mEER *PGK:CD63-eGFP* cells showed high levels of eGFP fluorescence. eGFP fluorescence was also detected in lower levels in modified mEER *ALDH1A1:CD63-eGFP* and *SRE:CD63-eGFP* cells, suggesting that eGFP brightest cells may constitute the CSC population (**Fig 1E**). To test this hypothesis, we flow sorted the top 5% of the engineered cells based on eGFP intensity and evaluated their expression of stemness markers and ability to form orospheres in low-attachment culture. RT-qPCR assay revealed that both mEER *ALDH1A1:CD63-eGFP* bright and mEER *SRE:CD63-eGFP* bright cells showed significantly higher expression levels of the stemness markers *ALDH1A1*, *Nanog* and *SOX2* when compared to eGFP- cells (**Fig 1F**). Similarly, MOC2 *ALDH1A1:CD63-eGFP* bright and MOC2 *SRE:CD63-eGFP* bright cells showed significantly greater expression of the stemness marker *ALDH1A1* than eGFP- cells (**S3C Fig**). Importantly, flow sorted mEER *ALDH1A1:CD63*-eGFP bright cells efficiently formed orospheres when cultured in serum-free Low-Attachment (LA) conditions while eGFP- cells were not able to form orospheres but showed small cellular aggregations (**Fig 1G**). Similarly, MOC2 *ALDH1A1:CD63-eGFP* bright cells formed big cell clusters when cultured in LA conditions (**S3D Fig**). To confirm that the orospheres we obtained in culture contain *bona fide* CSC, we performed gene expression analysis on mEER *ALDH1A1:CD63-eGFP* bright orospheres and found that they expressed significant higher levels of the stemness markers *ALDH1A1*, *Nanog*, *Oct-4, CD-133 and Sox-2* than unsorted mEER *ALDH1A1:CD63-eGFP* cells cultured in attachment conditions (**Fig 1H**). Additionally, the *in vivo* expression of the HNSCC CSC marker SOX2 was analyzed in mice carrying tumors formed by MOC2 *SRE:CD63-eGFP* cells. Expectedly, only MOC2 eGFP+ cells also co- expressed SOX2 while no SOX2 expression was found in eGFP- tumor cells (**S3E Fig**). Altogether, these data confirm the ability of *ALDH1A1: CD63-eGFP* and *SRE:CD63-eGFP* expression cassettes to restrict expression of the tEV reporter within the CSC-enriched subpopulation of mEER and MOC2 cancer cell lines.

## tEVs^csc preferentially target MHC-II–macrophages and PD-1+ T cells

We and others have previously investigated the interactions that occur in the TME between tEVs and immune cells [42–49]. Mononuclear phagocytes and tumor endothelial cells were

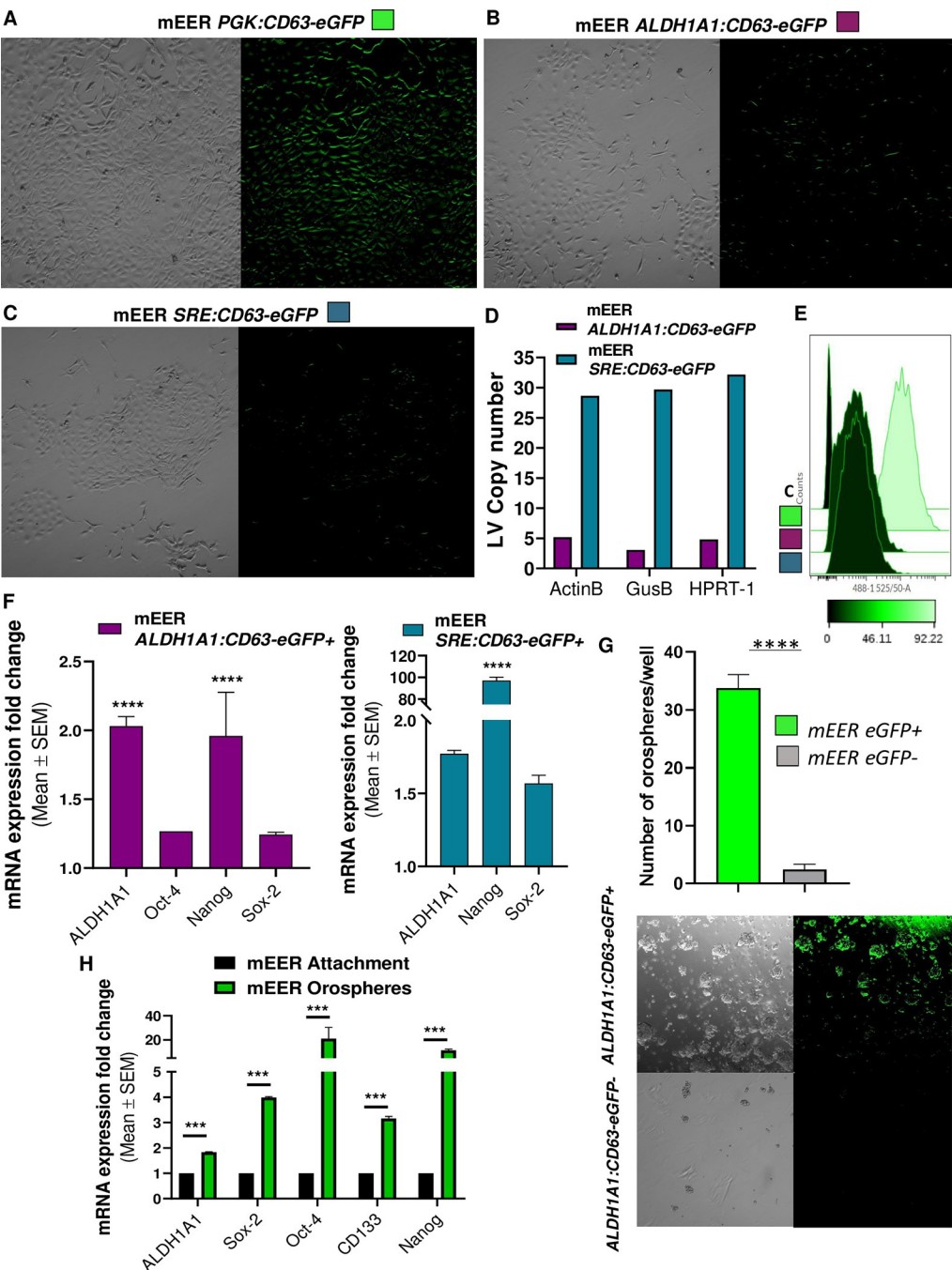

**Fig 1. OSCC mEER CSC model characterization.** (**A**) Representative fluorescence confocal microscopy images of in vitro cultured mEER cells stably transduced with *PGK:CD63-eGFP* reporter. Green cells represent the total tumor cells. (**B, C**) Representative fluorescence confocal microscopy images of cultured mEER cells carrying the *ALDH1A1: CD63-eGFP* or the *SRE:CD63-eGFP* expression cassettes. Green cells represent the CSC population. (**D**) Graph representing the LV copy number present in genetically modified *ALDH1A1:CD63-eGFP* and *SRE:CD63-eGFP* mEER cells assessed by PCR analysis. (**E**) Flow cytometry analysis displaying eGFP fluorescence in unmodified (Control, C) and genetically modified *PGK:CD63-eGFP* (green), *ALDH1A1:CD63-eGFP* (purple) and *SRE:CD63-eGFP* (teal) mEER cells. (**F**) Relative increase in stemness gene expression of flow sorted brightest mEER eGFP + cells compared to eGFP-cell population analyzed by RT-qPCR. (**G**) Graph depicting the number of orospheres per well generated from flow sorted brightest mEER eGFP + cells versus eGFP- cells. Representative microscopy images of orospheres growing from flow sorted *ALDH1A1:CD63-eGFP*+ mEER and *ALDH1A1:CD63-eGFP*- mEER cells cultured in 3D tumorsphere-specific medium are displayed. Green fluorescence indicates the high content of CSC in the orospheres. (**H**) RT-qPCR analysis showing the stemness gene expression signature of mEER orospheres enriched in ALDH1A1:CD63-eGFP

+ CSC compared to unmodified mEER cells growing in attachment conditions assessed by RT-qPCR. Holm-Sidak´s t test was used for statistical analysis.

among the cell types that bound tEVs at the highest rate [49]. Whether tEVs$^{CSC}$ possess a distinct tropism toward tumor infiltrating immune cells is still unknown. In order to test if tEVs$^{CSC}$ preferentially interact with specific immune cell subsets, we challenged mice with mEER tumor cells carrying the *ALDH1A1*:*CD63-eGFP* or the *SRE*:*CD63-eGFP* expression cassette. As control, we used mice bearing mEER tumor cells carrying the *PGK*:*CD63-eGFP* expression cassette. Flow cytometry-based analysis of tumors revealed the presence of different levels of CD63-eGFP+ cells among groups (**Fig 2A**). We then asked if differences in tEVs and tEVs$^{CSC}$ tropism exist within functional subsets of CD45+ cells. Specifically, F4/80+ MHCII + and F4/80+ MHCII- Macrophages (Mac), inflammatory and resident monocytes (Mo), PD-1+ and PD-1- T cells, Neutrophils (Neu), B cells and dendritic cells (DC) immune subpopulations were analyzed (**S4 Fig**). In tumors formed by mEER cells constitutively expressing CD63-eGFP, CD45+ CD63-eGFP+ cells were composed mainly of MHC-II+ Mac (30.6%), B cells (16.5%) and Inflammatory Mo (16%), followed by Neu (12.9%) (**Fig 2B**). When we analyzed tumors expressing either of the tEVs$^{CSC}$ reporters, we observed an increased fraction of CD63-eGFP+ MHC-II–Mac among CD45+ CD63-eGFP+ cells in tEVs$^{CSC}$ reporter tumors (27.8%, average between *ALDH1A1*:*CD63-eGFP* and *SRE*:*CD63-eGFP*) compared to *PGK*: *CD63-eGFP* tumors (5.7%). Interestingly, we observed an enrichment in the interactions between mEER tEVs$^{CSC}$ and PD-1+ T cells (11.4%), as compared to tEVs (3.4%). (**Fig 2C**). We observed statistically significant differences between the percentage of CD63-eGFP+ MHC-II + Mac infiltrating tumors constitutively expressing the tEV reporter (87.4%) and those present in tumors carrying CSC reporters (10.5%), indicating that MHC-II+ Mac predominantly uptake non-CSC tEVs. On the other hand, the percentage of CD63-eGFP+ MHC-II–Mac did not significantly change, suggesting that MHC-II–Mac predominantly uptake tEVs$^{CSC}$ (**Fig 2D**). When we analyzed monocyte subsets, we observed a significant decrease in the percentage of CD63-eGFP+ monocytes from both inflammatory and resident subsets, indicating that monocytes predominantly uptake non-CSC tEVs. Similar results were observed with Neu, B cells and DC subsets, indicating that those populations preferably uptake non-CSC tEVs (**Fig 2D**). By labeling T cells with the activation marker PD-1, we observed that the percentage of CD63-eGFP+ PD-1+ T cells did not significantly change between tumors constitutively expressing the tEVs reporter (9.6%) and those present in tumors carrying CSC reporters (6%), suggesting that, similarly to MHC-II–Mac, also PD-1+ T cells predominantly uptake tEVs$^{CSC}$ (**Fig 2C**). To highlight these differences, we calculated a tEVs$^{CSC}$ specificity index by dividing the percentage of CD45+ CD63-eGFP+ immune cells for each subset in the tEVs$^{CSC}$ groups by the percentage of the corresponding CD45+ CD63-eGFP+ subsets from the tEVs group. We observed that the index for MHC-II- Mac and PD-1+ T cells was significantly increased for those populations when compared to the index of all the other immune cell subsets. While tEVs$^{CSC}$ specificity index mean was 0.87 for MHC-II–Mac and 0.63 for PD-1+ T cells, all other tested subsets were below 0.21 (**Fig 2E**). Altogether, these data indicate that tEVs$^{CSC}$ possess a preferential tropism towards MHC-II–Mac and PD-1+ T cells.

## Cancer stem cells and macrophages share the same niche within the tumor microenvironment

Our results so far indicate that MHC-II–Mac are selectively binding tEVs$^{CSC}$. We next aimed to identify the mechanisms of such preferential binding. We hypothesized that CSC and MHC-II–Mac may share the same niches within the tumor microenvironment, which would

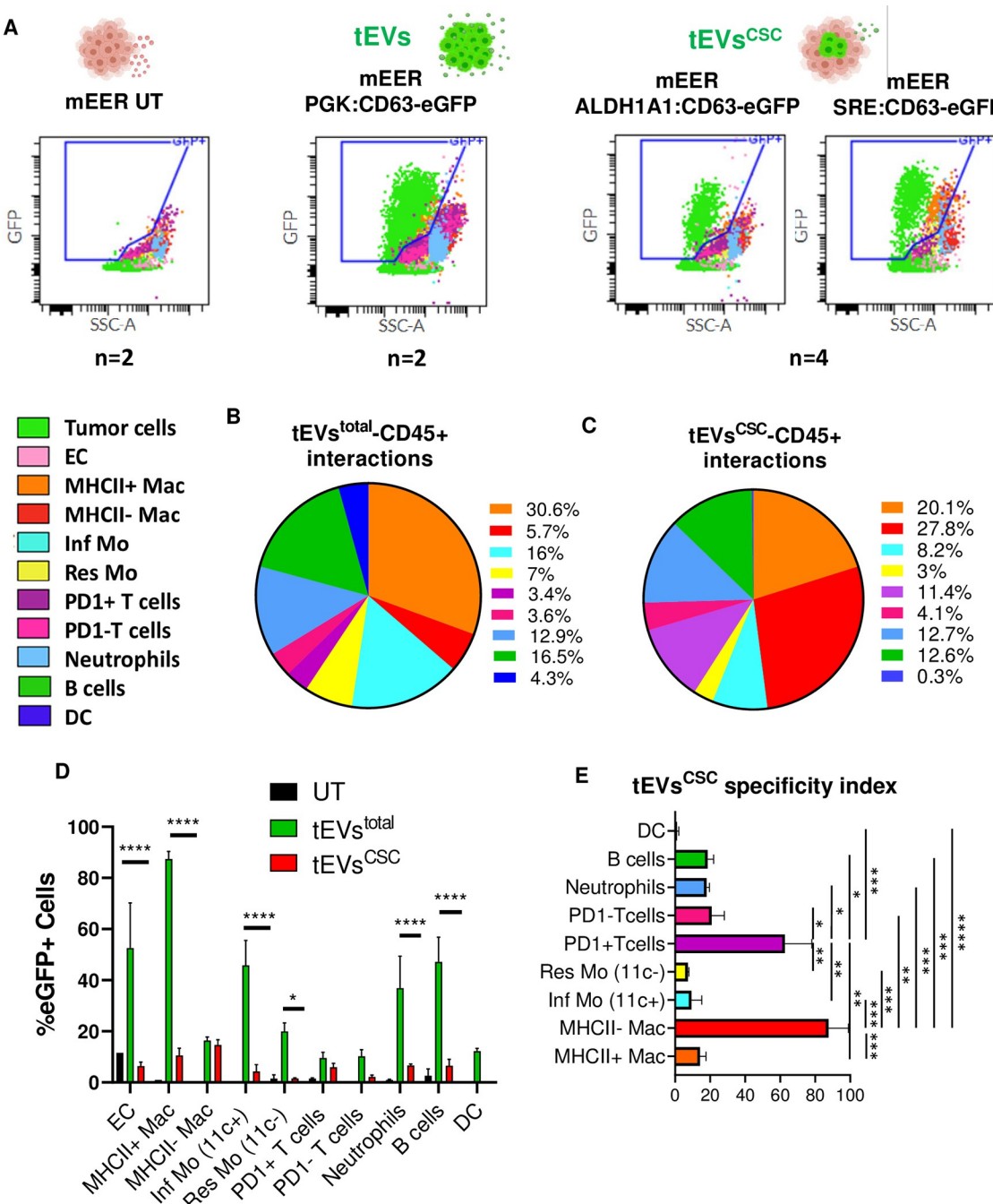

**Fig 2. *In vivo* released mEER tEVs<sup>CSC</sup> target MHC-II- Mac and PD-1+ T cells in the TME. (A)** Representative overlaid plot of color-coded cell subsets present in tumors recovered from mice bearing genetically modified mEER cells studied by flow cytometry. Three different experimental tumor groups were analyzed: tumors formed by i) unmodified mEER cells ii) mEER cells modified with *PGK:CD63-eGFP* construct (labeling tEVs) and iii) mEER cells engineered with *ALDH1A1:CD63-eGFP* and *SRE:CD63-eGFP* LVs (labeling tEVs<sup>CSC</sup>). Sample number (n) for each group is indicated below each plot. CD63-eGFP+ gate is indicated in each case. **(B)** Representative graph of the % of specific tEVs–CD45+ immune cell subsets interactions from all eGFP+ events in tumors CD45 + cells, showing a major interaction between tEVs and MHC-II+ Mac. **(C)** Representative graph of the % of specific tEVs<sup>CSC</sup>–CD45 + immune cell subsets interactions from all eGFP+ events in tumors CD45+ cells, showing a major interaction between tEVs and MHC-II- Mac. **(D)** Summary graph showing the % of cell subsets presenting CD63-eGFP+ fluorescence in unlabeled EVs tumors (UT), tEV-labeled tumors and tEV<sup>CSC</sup>-labeled tumors. **(E)** tEV<sup>CSC</sup> specificity index showing the preferential interaction of secreted tEVs<sup>CSC</sup> with specific immune cell subsets. Two-way ANOVA Tukey's and Holm-Sidak's multiple comparisons tests were used for statistical analysis.

increase exposure to tEVs$^{CSC}$. We tested this hypothesis by imaging tumor sections. To this end, we performed immunofluorescence staining for CD45 and F4/80 on tumor sections from mice carrying either *PGK*:*CD63-eGFP* or *ALDH1A1*:*CD63-eGFP* expressing tumors. As expected, confocal microscopy images showed a broad CD63-eGFP+ signal in *PGK*: *CD63-eGFP* tumors (**Fig 3A**, **S5A Fig**), whereas fewer CD63-eGFP+ cells were found in *ALDH1A1*:*CD63-eGFP* tumors (**Fig 3B**, **S5B Fig**). Among CD45+ F480+ tumor macrophages, many were found at the tumor periphery. When considering those infiltrating the tumor mass, we observed a significant association of CD45+ F480+ tumor Mac with CSC, as quantified by measuring the distance between them in three distinct areas from 5 random tumor sections (**Fig 3C**). These results suggest that the location of tumor macrophages may favor their preferential uptake of tEVs$^{CSC}$.

## Location-dependent labeling uncovers short-range interactions between CSC, tEVs$^{csc}$ and MHC-II–Mac, PD-1+ T cells in the TME

To further investigate which immune cells are part of the CSC niche, we employed a location-dependent labeling strategy we recently validated [50]. The approach takes advantage of a membrane-bound bacterial transpeptidase, Sortase A (SrtA), which can catalyze the contact dependent transfer of a fluorescent reporter. We previously optimized and validated SrtA design, reaction requirements and efficiency *in vitro* [50]. The reporter is monomeric Scarlett fluorescent protein (mSca) fused with SrtA recognition sequence (LPETGG) and with a secretory signal sequence, which leads to the secretion of the SrtA substrate in the extracellular environment. SrtA is engineered to be present on the outer side of cellular membranes. Of note, mSca is ubiquitously expressed by all tumor cells under the control of a constitutive bi-directional promoter, along with CD63-eGFP, whereas SrtA is specifically expressed by CSC under the regulation of the *ALDH1A1* promoter (**S1C and S2A Figs**) [51]. This experimental design allows us to detect both CSC- and tEVs$^{CSC}$-mediated labeling of immune cells recruited to the CSC niche based on mSCA fluorescence (**Fig 4Ai**). A detailed scheme of the experimental design is presented in **Fig 4Aii**. In order to control for baseline endocytosys/phagocytosis of mSca by immune cells, we included a control group in which CSC do not express SrtA while the bulk of tumor cells (including CSC) still secrete mSca. The values for SrtA activity are calculated by subtracting this baseline internalization of mSca.

Flow cytometric analysis revealed that, as expected, the immune cell subsets most frequently interacting with tumor cells (as measured by tEV uptake) were MHC-II+ Mac (21.3% of total CD45+ CD63-GFP+ cells), followed by B cells (20.6%) and Neu (13%) (**Fig 4C**), similarly to what we observed before (**Fig 2B**). When the same tumors were analyzed to identify immune cell subsets found within the CSC niche, we found that the highest fraction of CD45+ mSca + cells corresponded to MHC-II- Mac (30.5%), followed by MHC-II+ Mac (21.4%) and PD-1 + T cells (13%) (**Fig 4D**). When we analyzed tumor-infiltrating CD45+ cell subsets presenting eGFP and mSca fluorescence, we observed a significant lower percentage of mSca+ cells compared to eGFP+ cells in all immune subsets indicating that they predominantly uptake non-CSC tEVs. Interestingly, we observed that the percentage of CD63-eGFP+ and mSca+ cells did not significantly change in MHC-II- Mac and PD-1+ T cells, suggesting that these subsets were predominantly located within the CSC niche (**Fig 4E**). We then calculated a CSC niche-specificity index to summarize these results in one value. The CSC niche-specificity index increased for MHC-II- Mac (0.63) and PD-1+ T cell subpopulation (0.44), while it remained lower than 0.23 for the rest of the immune cell subsets (**Fig 4F**). These results are consistent with a model in which MHCII- Mac and PD1+ T cells dwell in the proximity of CSC and may influence each other.

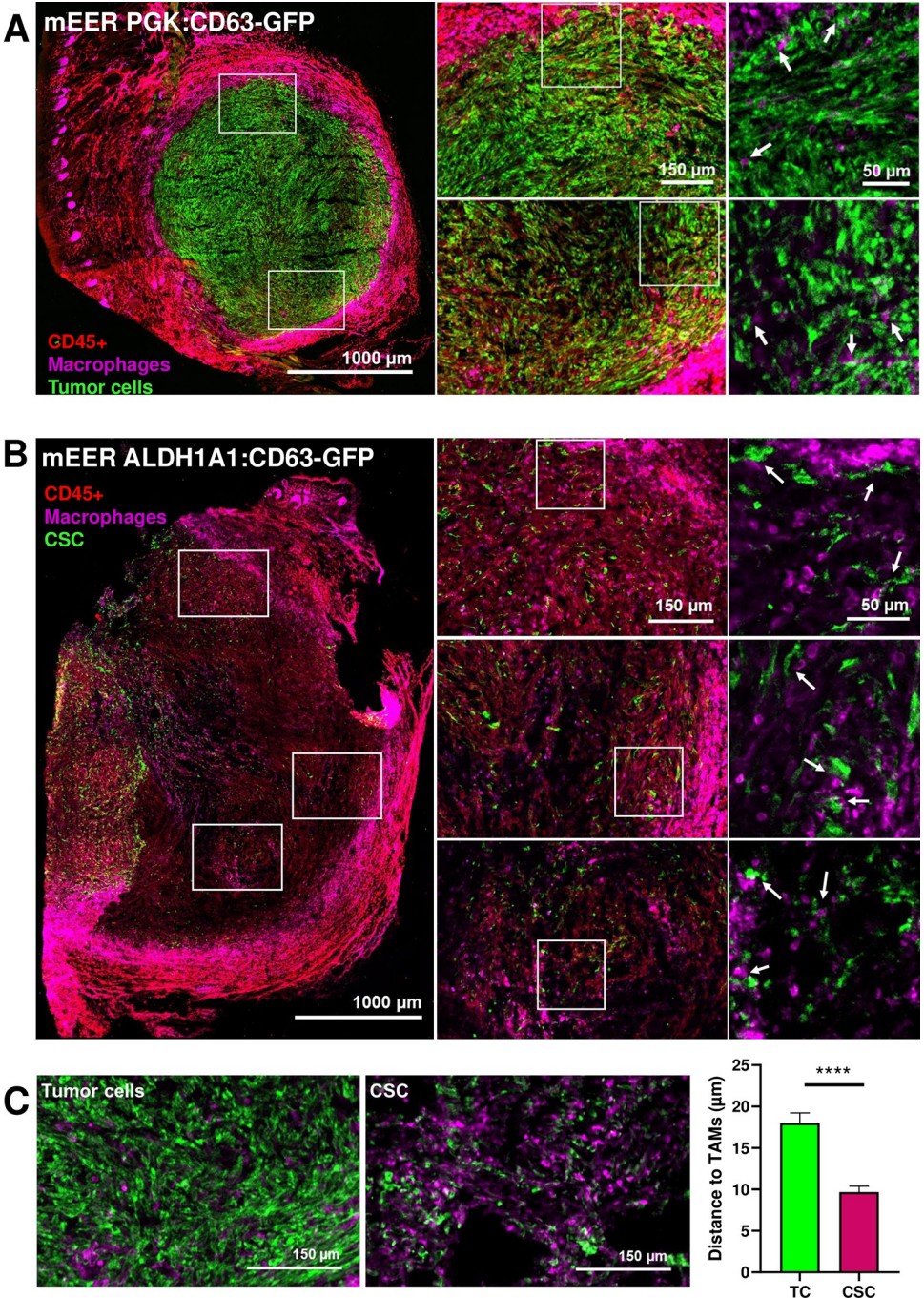

**Fig 3. CSC show close localization to TAMs in the TME. (A)** Representative IF images of tumor sections carrying mEER *PGK:CD63-eGFP* tumor cells. eGFP+ cells represent the total tumor cell population. Arrows in the insets indicate TAMs-tumor cells interactions. **(B)** Representative IF images of tumor sections presenting mEER *ALDH1A1: CD63-eGFP* tumor cells. eGFP+ cells represent the CSC population. Arrows in the insets indicate TAMs-CSC interactions. **(C)** Examples of analyzed tumor areas employed to measure the distance in μm observed between tumor eGFP+ cells, eGFP+ CSC and TAMs using ImageJ software. Mann-Whitney test was used for statistical analysis.

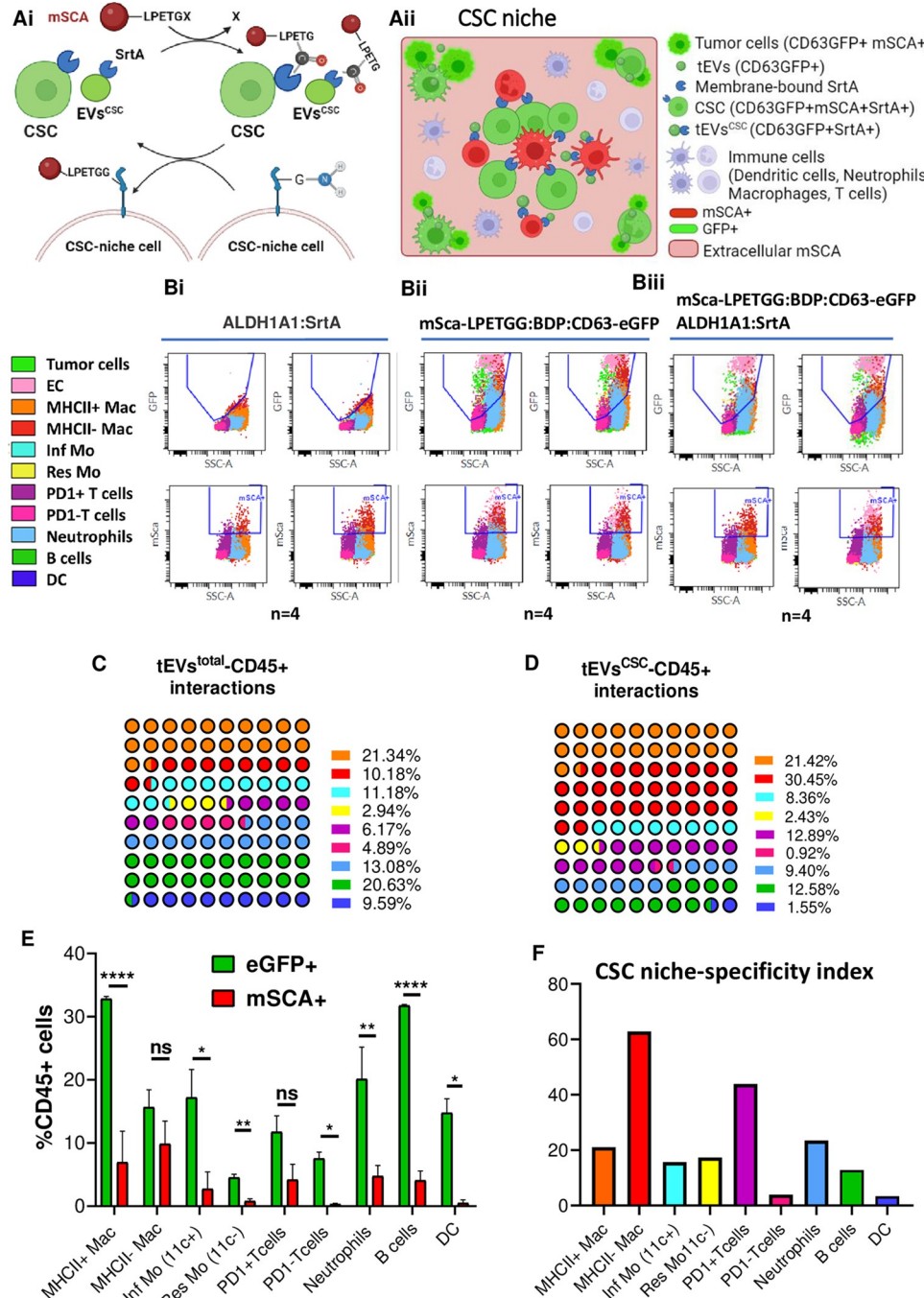

**Fig 4. Second-degree labeling via surface interactions reveals short range contacts between CSC, their tEVs$^{CSC}$ and MHC-II–Mac, PD-1+ T cells in the CSC niche. (Ai)** Illustrative schemes of SrtA enzymatic activity. SrtA present on the outer membranes of CSC and tEVs$^{CSC}$ catalyzes the transfer of the reporter protein mSca on CSC niche cells. SrtA recognizes the sequence LPETGG. **(Aii)** Scheme summarizing experimental design to uncover the immune component in the CSC niche. Since only CSC and tEVs$^{CSC}$ express SrtA, this strategy allows to label host cells with mSca reporter based on their proximity to CSC and their EVs. **(B)** Overlaid plot representation of different cell populations present in tumors from mice bearing genetically modified mEER cells analyzed by flow cytometry. Two plots from n = 4 are presented for each group. **(C)** Representative graphs showing the % of CD63-eGFP+ immune cells (that is, immune cells binding to tEVs) in analyzed tumors for each of the analyzed subsets with respect to all CD63-eGFP+ CD45+ immune cells. **(D)** Representative graphs showing the % of mSca+ immune cells (that is, immune cells in the proximity of CSC and their tEVs$^{CSC}$) for each of the analyzed subsets with respect to all mSca + CD45+ immune cells. **(E)** Summary graph presenting the % of immune cell subsets presenting CD63-eGFP+ and

mSca+ fluorescence in tumors carrying modified mEER *ALDH1A1:SrtA/mSca-LPETGG:BDP:CD63-eGFP* tumor cells test. Percentage of CD63-eGFP+ cells for each immune subset represents the % of cells that had interacted with tEVs. mSca+ cells for each immune subset represent the % of cells within the CSC niche. **(F)** CSC niche-specificity index indicating the % from tEVs[total]-immune cell subsets interactions that correspond to specific CSC niche-immune cell subsets interactions. Two-way ANOVA Holm-Sidak´s multiple comparison test and multiple t tests were used for statistical analysis. BDP: Bi-directional promoter.

## Discussion

Tumor released EVs are key modulators of tumor immunity [30, 42–49]. The population of cancer cells is very heterogeneous, with some clones retaining stronger stem-like activity [14–19, 52, 53]. Distinguishing between the contribution of CSC from that of more differentiated cancer cells to the total tEV pool has been very challenging, especially *in vivo*. A recent study used genetic labeling of tEV-targeted cells and focused on tumor cell-to-tumor cell signaling via EVs [54]. The Authors uncovered an effective cooperation network mediated by tEVs and led by CSC, suggesting that a similar network may be in place between CSC and immune cells. Given the importance of CSC in cancer biology, new technologies that allow to dissect the influence of native tEVs[CSC] are necessary. In this study, we present a novel strategy to effectively track CSC-derived EVs in the tumor microenvironment (TME) under physiological conditions, avoiding any *in vitro* manipulation of EVs. Here, we demonstrate proof-of-concept studies using genetic labeling of tEVs[CSC] to identify targeted cells in the TME, with a specific focus on immune cell populations. We observed a surprising selectivity of tEVs[CSC] in targeting specific immune cell subsets, namely MHC-II–Mac and PD-1+ T cells, which were also found enriched within the CSC niche.

Analysis of tumors with fluorescently labeled tEVs and tEVs[CSC] showed that the fraction of immune cell subsets presenting CD63-eGFP fluorescence was higher in the tEVs labeled group than in the tEVs[CSC] labeled group. These results are expected since tEVs[CSC] represent a minor fraction of bulk total tEVs. Deeper analysis of CD63-eGFP+ immune cell subpopulations in tumors identified Mac as the immune cell type with the highest interaction rate with both tEVs and tEVs[CSC]. These results are also expected, as TAMs constitute the most abundant population of tumor-infiltrating immune cells in TME [55]. TAMs are educated by environmental factors to exhibit a spectrum of polarization phenotypes usually associated with specific functional states. One key functional biomarker of TAM polarization is MHC-II [56]. Among the roles ascribed to MHC-II–Mac within the TME are immunosuppression [57, 58], lymph/angiogenesis [59], ECM deposition [60, 61] and metastasis [62]. Here, we report a clear difference between the interaction rates of MHC-II+ and MHC-II- Mac populations with each EVs fraction. While tEVs mainly interacted with MHC-II+ Mac, tEVs[CSC] showed a significant preference toward MHC-II- Mac, as quantified by the specific interaction index. Numerous studies have shown the ability of tEVs to polarize TAMs towards pro-tumorigenic MHC-II- phenotypes [63–66]. In this respect, our data suggest that tEVs[CSC] may be the main tEV subset responsible for the reported TAMs polarization.

Our findings provide a potential mechanistic explanation of the recently reported maintenance of CSC niche by MHC-II–Mac [26, 28, 67]. The CSC niche is particularly important to support CSC self-renewal, repopulation potential, and tumor initiation [68]. CSC contribute to the creation of a niche by inducing Mac polarization towards an immunosuppressive phenotype (MHC-II–), which in turn promotes and supports CSC aggressiveness [69–72]. The relevance of TAMs in CSC biology is reinforced by a growing list of TAM-derived factors, including IL -6, IL-8, and CXCL1, that have been implicated in the maintenance of CSC stemness in different types of cancer [69–74]. Here, we report that tEVs[CSC] as an additional mediator targeting MHC-II–Mac. In addition, novel location-dependent labeling approaches based

on surface expression of Sortase-A demonstrated that CSC and MHC-II–Mac share the same niche. As the presence of MHC-II–Mac and CSC populations in human tumors has been correlated with a poor prognosis for many types of cancer [57, 75, 76], deeper knowledge of this communication network will be important to identify novel therapeutic opportunities.

Together with MHC-II–Mac, PD-1+ T cells also displayed specific interaction rates with tEVs$^{CSC}$, as compared to total tEVs. The PD-1/PD-L1 pathway is a key immunosuppressive mechanism with significant clinical implications in many solid cancer types, including HNSCC [77, 78]. Tumor EVs can also present PD-L1 on their surface, playing critical immunosuppressive roles when binding to PD-1+ T cells [79, 80]. Specifically, circulating PD-L1$^{high}$ exosomes in HNCC patients' plasma–but not soluble PD-L1 levels, have been associated with disease progression [81]. Our study reveals that tEVs$^{CSC}$ specifically interacted with PD-1+ T cell subsets, which may suggest the presence of PD-L1 ligand on tEVs$^{CSC}$. We also show that PD-1+ T cells and CSC shared the same niche. Thus, it is conceivable that tEVs$^{CSC}$ may be primarily responsible for competing with immune checkpoint inhibitors in the clinic. For this reason, identification of the specific immune-modulators present on tEVs$^{CSC}$ is needed to specifically target their immunosuppressive signals.

To the best of our knowledge, this work has two main limitations. First, although we are showing that generic transmembrane domains (such as that we used to tether SrtA to cell membranes) lead to labeling of EV subpopulations expressing both CD63 and CD81, we cannot exclude that other EV subsets (such as AnnexinA1+ and ARRDC1+ shedding microvesicles [82] may target different immune cells. This concern is mitigated by the fact that EV subsets are often overlapping in biomarkers [82], and by the yet unknown *in vivo* abundance of shedding microvesicles, as compared to that of multivesicular body-derived EVs. The second limitation derives from the fact that current genetic technology cannot target a reporter only to the EV compartment while avoiding expression in the parental cell which translates the reporter itself and produces the EVs.

In conclusion, the present work not only establishes a novel technological platform to study tEVs$^{CSC}$ and their roles in the TME at the single cell level, but also identifies specific immune cell subsets linked to CSC biology. A better understanding of these microanatomical crosstalks will lead to a better knowledge of the differential effect of tEVs$^{CSC}$ on tumor progression.

## Supporting information

**S1 Fig. Schematic of the lentiviral vectors (LV). (A)** *dLNGFR*:*mCMV-PGK*:*CD63-eGFP* lentivector[49]. **(B)** *ALDH1A1*:*CD63-eGFP* and *SRE*:*CD63-eGFP* lentivectors. **(C)** *ALDH1A1*:*SrtA* and *SS-mSca-LPETGG*:*mCMV-PGK*:*CD63-eGFP* lentivectors.
(PDF)

**S2 Fig. Flow cytometric validation of EV reporters. (A)** mEER cells expressing mSca and SrtA (middle) or mSca and CD63-eGFP (right). Control untransduced cells are on the left. SrtA is detected by the Flag peptide. Note that these are constitutive expression vectors, not CSC specific. **(B-C)** Nano-flow analysis of EV preparations labeled with two fluorescent antibodies, Flag and CD63 (B) or Flag and CD81 (C). Triton treatment (top) confirms the vesicular nature of the signal.
(PDF)

**S3 Fig. OSCC MOC2 CSC model. (A, B)** Representative fluorescence microscopy images of cultured MOC2 *ALDH1A1*:*CD63-eGFP* cells and *SRE*:*CD63-eGFP* cells in culture. **(C)** Relative increase in stemness gene expression of flow sorted MOC2 eGFP + cells compared to eGFP-cells analyzed by RT-qPCR. **(D)** Representative images of flow sorted MOC2 *ALDH1A1*:

*CD63-eGFP+* cells growing in 3D tumorspheres specific medium. **(E)** Representative IF images of tumor sections presenting MOC2 *SRE:CD63-eGFP* tumor cells. Green cells represent the CSC population. Co-expression of CSC reporters SOX2 and eGFP are found.
(PDF)

**S4 Fig. Flow cytometry gating analysis of studied tumors.** Example of the gating strategy used to characterize the immune cells subsets present in analyzed tumors of *in vivo* experiments using Cytobank software.
(PDF)

**S5 Fig.** Additional representative IF images of tumor sections carrying mEER *PGK*: *CD63-eGFP+* (**A**) and mEER *ALDH1A1:CD63-eGFP+* (**B**) tumor cells.
(PDF)

**S1 Table. Taqman probes references (Thermo Fisher Scientific).**
(DOCX)

## Author Contributions

**Conceptualization:** Patricia Gonzalez-Callejo, Ferdinando Pucci.

**Data curation:** Patricia Gonzalez-Callejo.

**Formal analysis:** Patricia Gonzalez-Callejo.

**Funding acquisition:** Patricia Gonzalez-Callejo, Ferdinando Pucci.

**Investigation:** Patricia Gonzalez-Callejo, Zihan Guo, Tahereh Ziglari, Natalie Marcia Claudio, Kayla Hoang Nguyen.

**Methodology:** Patricia Gonzalez-Callejo, Joaquim Seras-Franzoso, Ferdinando Pucci.

**Project administration:** Ferdinando Pucci.

**Resources:** Naoki Oshimori, Ferdinando Pucci.

**Software:** Patricia Gonzalez-Callejo.

**Supervision:** Ferdinando Pucci.

**Validation:** Ferdinando Pucci.

**Visualization:** Patricia Gonzalez-Callejo.

**Writing – original draft:** Patricia Gonzalez-Callejo, Joaquim Seras-Franzoso, Ferdinando Pucci.

**Writing – review & editing:** Patricia Gonzalez-Callejo, Naoki Oshimori, Ferdinando Pucci.

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
