## [Decision Letter · Decision Letter 0]

17 Aug 2022

PONE-D-22-21278Cancer stem cell-derived extracellular vesicles preferentially target MHC-II– macrophages and PD1+ T cells in the tumor microenvironmentPLOS ONE

Dear Dr. Pucci,

Thank you for submitting your manuscript to PLOS ONE. After careful consideration, we feel that it has merit but does not fully meet PLOS ONE’s publication criteria as it currently stands. Therefore, we invite you to submit a revised version of the manuscript that addresses the points raised during the review process.

Please address the comments from the reviewer, as well as the comment from the editor listed below:1. Figure 4Aii, immune cells are depicted in the schematic presentation, but not actually in the cartoon presented. Please double check. 

We look forward to receiving your revised manuscript.

Kind regards,

Daotai Nie, Ph.D.

Academic Editor

PLOS ONE

Journal Requirements:

2. To comply with PLOS ONE submissions requirements, in your Methods section, please provide additional information regarding the experiments involving animals and ensure you have included details on (1) methods of sacrifice, (2) methods of anesthesia and/or analgesia, (3) efforts to alleviate suffering, and (4) maximum tumour volume allowed and the method by which tumour volumes were calculated.

Reviewers' comments:

Reviewer's Responses to Questions

**Comments to the Author**

1. Is the manuscript technically sound, and do the data support the conclusions?

Reviewer #1: Yes

2. Has the statistical analysis been performed appropriately and rigorously? 

Reviewer #1: No

3. Have the authors made all data underlying the findings in their manuscript fully available?

Reviewer #1: Yes

4. Is the manuscript presented in an intelligible fashion and written in standard English?

Reviewer #1: Yes

5. Review Comments to the Author

Reviewer #1: This work conducted by the Pucci Group suggests that EVs from different cell-of-origins within tumors, CSC defined by ALDH1A1 or SRE in this case, contribute to the heterogenicity of immune TME. The observation is interesting, the technical approaches are novel, the manuscript is well-written, and the information about experimental methods are adequate. However, addressing some points can strengthen this study and further support the conclusions.

1. The definition of CSCs by single or two markers remains controversial. Are ALDH1A+ or SRE positive cells accountable for the "cancer sternness" in HNSCC, or simply reflecting a specific and small fractions of heterogenous CSCs? That being said, indicating the orosphere forming efficiency of sorted ALDH1A1:CD63-eGFP+ mEER cells (Fig 1G) can partially answer the question.

2. The measurement of the distance between tumor cell-TMA in Fig 3 should be further clarified. How many samples were included in the experiments, and whether the assay was conduced in a double-blind manner? In addition, are ALDA1A1+ cells in the tumors derived from total tumor population (Fig 3A) closer to TAMs? A co-staining method can help to address this. This should be important since the interactions could be altered within a heterogenous intratumoral environment.

3. It should be clarified whether the attraction of TAMs or other immune cells are directly caused by EVs. More specifically, does neutralizing CD63 by antibody block the recruitment of TMAs to the peripheral or intratumoral areas?

6. PLOS authors have the option to publish the peer review history of their article (what does this mean?). If published, this will include your full peer review and any attached files.

Reviewer #1: No

---

## [Author Response · Author response to Decision Letter 0]

5 Oct 2022

We would like to thank the Editor and the Reviewer for taking the time to read our work and to appreciate its value to the CSC and EV community. Please find below in navy font a point-by-point response to their comments.

Editor:

Figure 4Aii, immune cells are depicted in the schematic presentation, but not actually in the cartoon presented. Please double check. 

We agree that the schematic was misleading by not showing unlabeled immune cells (purple), that we now added along with labeled immune cells (red) within the CSC nice and immune cells labeled by total tumor cells EVs (green).

Reviewer:

The definition of CSCs by single or two markers remains controversial. Are ALDH1A+ or SRE positive cells accountable for the "cancer sternness" in HNSCC, or simply reflecting a specific and small fractions of heterogenous CSCs? That being said, indicating the orosphere forming efficiency of sorted ALDH1A1:CD63-eGFP+ mEER cells (Fig 1G) can partially answer the question.

We appreciate the Reviewer’s request to add the orosphere forming efficiency, which is an important parameter for relating our work to others in the literature. We have now added the assessed orosphere forming efficiency (Fig. 1G), which is 0.66%.

The measurement of the distance between tumor cell-TMA in Fig 3 should be further clarified. How many samples were included in the experiments, and whether the assay was conduced in a double-blind manner?

Thank you for noticing this missing detail in the methods. We have added that the distance measurement was made blindly by 3 independent observers. This distance was quantitatively assessed in 5 random tumor sections per mice group. These details have now been included in the methods section. Representative images were displayed in Figure 3.

In addition, are ALDA1A1+ cells in the tumors derived from total tumor population (Fig 3A) closer to TAMs? A co-staining method can help to address this. This should be important since the interactions could be altered within a heterogenous intratumoral environment.

We appreciate this Reviewer’s knowledge of the controversial issues present in the field of cancer stemness. From the Reviewer’s comment, we understand that the concern is whether our reporters label the majority the heterogeneous CSC populations or a specific subset. We have now performed the suggested experiment and co-stained with Sox2, a major CSC reporter in squamous carcinoma (PMID: 27634934). We observed a major co-localization between our reporters and Sox2 (Fig. S3E), which indicates that we effectively tagged heterogeneous CSC subpopulations. These results, the orosphere formation efficiency (Fig. 1G) and the fact that the ALDH1A1:eGFP+ and SRE:eGFP+ CSC populations express other stemness markers (Fig. 1 and S3), together indicate that our conclusions may be broadly applicable to most CSC subsets. 

It should be clarified whether the attraction of TAMs or other immune cells are directly caused by EVs. More specifically, does neutralizing CD63 by antibody block the recruitment of TMAs to the peripheral or intratumoral areas?

We agree with the Reviewer that understanding the mechanism of immune cell recruitment into the CSC niche is of interest. However, this goes beyond the scope of the manuscript under consideration, which is mainly focused on the characterization of the composition of the CSC niche in vivo. In addition, anti-CD63 antibodies have not been shown to deplete EVs, and likely will have the opposite effect on macrophages since opsonizing EVs with antibodies will increase their binding to macrophage Fc receptors. We hope the Reviewer will understand that answering this important question will be the focus of future work.

---

## [Decision Letter · Decision Letter 1]

6 Dec 2022

Cancer stem cell-derived extracellular vesicles preferentially target MHC-II– macrophages and PD1+ T cells in the tumor microenvironment

PONE-D-22-21278R1

Dear Dr. Pucci,

We’re pleased to inform you that your manuscript has been judged scientifically suitable for publication and will be formally accepted for publication once it meets all outstanding technical requirements.

Kind regards,

Daotai Nie, Ph.D.

Academic Editor

PLOS ONE

Additional Editor Comments (optional):

Reviewers' comments:

Reviewer's Responses to Questions

**Comments to the Author**

1. If the authors have adequately addressed your comments raised in a previous round of review and you feel that this manuscript is now acceptable for publication, you may indicate that here to bypass the “Comments to the Author” section, enter your conflict of interest statement in the “Confidential to Editor” section, and submit your "Accept" recommendation.

Reviewer #1: All comments have been addressed

2. Is the manuscript technically sound, and do the data support the conclusions?

Reviewer #1: Yes

3. Has the statistical analysis been performed appropriately and rigorously? 

Reviewer #1: Yes

4. Have the authors made all data underlying the findings in their manuscript fully available?

Reviewer #1: Yes

5. Is the manuscript presented in an intelligible fashion and written in standard English?

Reviewer #1: Yes

6. Review Comments to the Author

Reviewer #1: All concerns have been adequately addressed, and the manuscript is well written. I recommend to accept and publish this study.

7. PLOS authors have the option to publish the peer review history of their article (what does this mean?). If published, this will include your full peer review and any attached files.

Reviewer #1: No

---

## [Editor Report · Acceptance letter]

24 Jan 2023

PONE-D-22-21278R1 

Cancer stem cell-derived extracellular vesicles preferentially target MHC-II– macrophages and PD1+ T cells in the tumor microenvironment 

Dear Dr. Pucci:

I'm pleased to inform you that your manuscript has been deemed suitable for publication in PLOS ONE. Congratulations! Your manuscript is now with our production department. 

Kind regards, 

on behalf of

Dr. Daotai Nie 

Academic Editor

PLOS ONE